# Hydrochemical Characteristics and Water Quality of Shallow Groundwater in Desert Area of Kunyu City, Southern Margin of Tarim Basin, China

Runchi Tang [1,*,†], Shuning Dong [1,2,†], Mengfei Zhang [3], Zhenfang Zhou [1,2], Chenghang Zhang [4], Pei Li [2] and Mengtong Bai [1]

1    College of Geology and Environment, Xi'an University of Science and Technology, Xi'an 710054, China; dongshuning@cctegxian.com (S.D.); zhouzhenfang@cctegxian.com (Z.Z.); baimt18391505165@163.com (M.B.)
2    CCTEG Xi'an Research Institute (Group) Co., Ltd., Xi'an 710077, China; lipei@cctegxian.com
3    Xinjiang Geological Survey Institute of Sinochem Geology and Mine Administration, Urumqi 830017, China; zmf18392362618@163.com
4    School of Earth Sciences and Engineering, Sun Yat-sen University, Zhuhai 519080, China; zhangchx65@mail2.sysu.edu.cn
*    Correspondence: trc1212@stu.xust.edu.cn
†    These authors contributed equally to this work.

**Abstract:** The Tarim Basin in Xinjiang is located in the northwest inland arid region of China, but research shows that the region is rich in groundwater resources. To understand the hydrochemical characteristics and water quality of shallow groundwater in the desert area of the southern margin of the Tarim Basin, the groundwater was systematically sampled and tested. The ion characteristics and evolution mechanism of groundwater were analyzed by mathematical statistics, Schukarev classification, Piper three-line diagram, Gibbs model and ion ratio. Water quality was evaluated by the water quality index method (WQI) and irrigation water suitability-related parameters. The results indicated that the dominant cation in the study area is $Na^+$, and the main dominant anions are $SO_4^{2-}$ and $Cl^-$. According to total dissolved solids (TDS), the groundwater mainly belongs to brackish water and semi-saline water. The hydrochemical chemistry types are mainly $Cl \cdot SO_4$-$Na \cdot Mg$ type, followed by $Cl$-$Na$ type, and the ion source is mainly the weathering and dissolution of evaporation rock, silicate and sulphate. The hydrochemical process is primarily controlled by evaporation concentration and rock weathering, and the cation exchange is weak. Furthermore, the WQI spatial distribution map shows that the groundwater in the middle of the study area is unsuitable for drinking and there are two areas with high WQI values greater than 500. In contrast, the good-excellent groundwater is scattered in the East. The groundwater generally has high to very high salinity, with significant changes in alkalinity. In addition, 54% of the water samples exceed the magnesium hazard (MH) limit. Therefore, certain measures should be taken before irrigation. This study has important implications for the rational development and reasonable utilization of local groundwater.

**Keywords:** desert area; groundwater; water quality; hydrochemical characteristics; water quality index

## 1. Introduction

As an indispensable source of water supply, groundwater is a vital strategic resource to support social and economic development and it plays a significant role in maintaining ecological equilibrium [1,2]. However, in the process of urbanization, industrialization and agricultural activities have a negative impact on groundwater quality. At the same time, excessive exploitation of groundwater leads to enormous pressures put on water supply demand, especially for developing countries such as China [3–5]. China has a shortage of water resources, with a low per capita share. On the other hand, major river basins and lakes are generally polluted, which further exacerbates the current situation of water shortage [6,7]. The severe water resource problem has become a hot topic of close

attention and an unavoidable century challenge in Chinese society [8]. Therefore, there is an urgent need to develop effective water resources management strategies and to strengthen research on the evolution mechanism of groundwater in order to ensure the sustainable development and utilization of water resources.

By analyzing the hydrochemical characteristics and influencing factors of groundwater, its formation and evolution pattern can be better revealed, which provides an essential basis for studying the origin of groundwater. Groundwater quality assessment is very helpful to understand its applicability, and is also necessary for its sustainable planning and management [9]. Among them, the water quality index (*WQI*) is not only an important parameter to evaluate whether the quality of drinking water meets the standard, but also a reference basis for the development, utilization and scientific allocation of water resources [10,11]. Previous researchers have conducted a large number of investigations and research on groundwater quality related issues: Chitsazan [12] investigated the groundwater chemical characteristics in the suburb of Urmia, Iran, and evaluated the quality of drinking water with an index model. It was found that the important processes to control the hydrochemical characteristics are mineral weathering, ion exchange, and human activities; affected by human activities, there are low-hazard pollutants in the suburbs. Tiwari et al. [13] conducted a qualitative evaluation of the mine water in the Sibocaro coalfield through the Piper diagram, saturation index (SI) and relevant parameters of irrigation water suitability. The results showed that the dissolution of minerals and ion exchange are the main processes to control the chemical change of water. The index value of most mine water samples does not exceed the standard, which is suitable for drinking and can be directly used for irrigation. To evaluate the genesis of water quality in Songyuan City, Northeast China, Yan et al. [14] measured the hydrogen and oxygen isotope composition and ion content of groundwater. The results show that the weathering and dissolution of silicate rock is the main factor affecting the ion composition of deep and shallow groundwater. At the same time, long-term artificial exploitation has caused the dispersion of pollutants in the shallow aquifer. In addition, scholars have conducted comprehensive evaluations of groundwater quality based on WQI, combined with multiple statistical analyses, fuzzy logic, geographic information systems, and other methods [15–19].

Due to the scarcity of surface water and atmospheric precipitation, groundwater has become the primary source of water supply in arid and semi-arid areas and is an essential factor affecting the sustainable development of these areas [20,21]. Therefore, scientific evaluation and sustainable development of groundwater are especially important in these areas [22]. The Tarim Basin in Xinjiang is a typical arid and semi-arid area with an arid climate and strong evaporation. Groundwater is an important water resource in this area. In recent years, there have been many studies on the hydrochemical characteristics of groundwater in the Tarim Basin [23–26]. However, these studies are primarily concentrated in the western and southern edge oasis areas of the Tarim Basin, lacking research on ionic characteristics and genetic mechanism of groundwater in desert areas. Furthermore, they mainly focus on the trace elements that affect human health in groundwater, and the comprehensive evaluation of groundwater quality is lacking. For this reason, this paper selects the Kunyu desert area in southern margin of the Tarim Basin as the study area. Combined with a large number of hydrological wells and hydrological borehole data, we analyze the hydrochemical types and causes of groundwater, and reveal the content changes and distribution laws of various chemical components of groundwater in the study area. At the same time, the water quality index method and relevant parameters of irrigation water suitability are used to evaluate water quality. It not only provides theoretical guidance for local groundwater protection and desertification prevention, but also provides a reference for the industrialization process of local reclamation agriculture.

## 2. Materials and Methods

### 2.1. Overview of the Research Area

Kunyu City is located in the southwest of Xinjiang Uygur Autonomous Region, at the northern foot of the Kunlun Mountains and the southern margin of the Tarim Basin. Based on the latest Gaofen-2 remote sensing image, the topography of the area was interpreted using remote sensing interpretation markers for each landform. Subsequently, GIS was used to calculate the proportion of each landform, and the interpretation results were verified through field investigations. The interpreted data was authentic and reliable. The mountainous area accounts for 33.3%, the Gobi desert accounts for 63%, and the oasis area only accounts for 3.7%, sporadically scattered among the Gobi desert. The study area is situated at the desert area of the lower Duwa River. The terrain is relatively open and flat, and the topography of small areas is relatively complex with many dunes. The topography is high in the west and low in the east, high in the south and low in the north, mainly developing mobile dunes, fixed-semi-fixed dunes and alluvial plain landforms. According to the information published by the local meteorological office, it belongs to a typical warm temperate inland arid climate. The main climatic characteristics are hot in summer, cold in winter, large diurnal temperature difference, scarce precipitation and strong evaporation. The average annual precipitation is 29 mm, and annual evaporation is 4410 mm. The Duwa River is the major surface water system flowing through the region, flowing from south to northeast, and gradually disappearing in the Gobi desert after passing through the Duwa Reservoir (Figure 1).

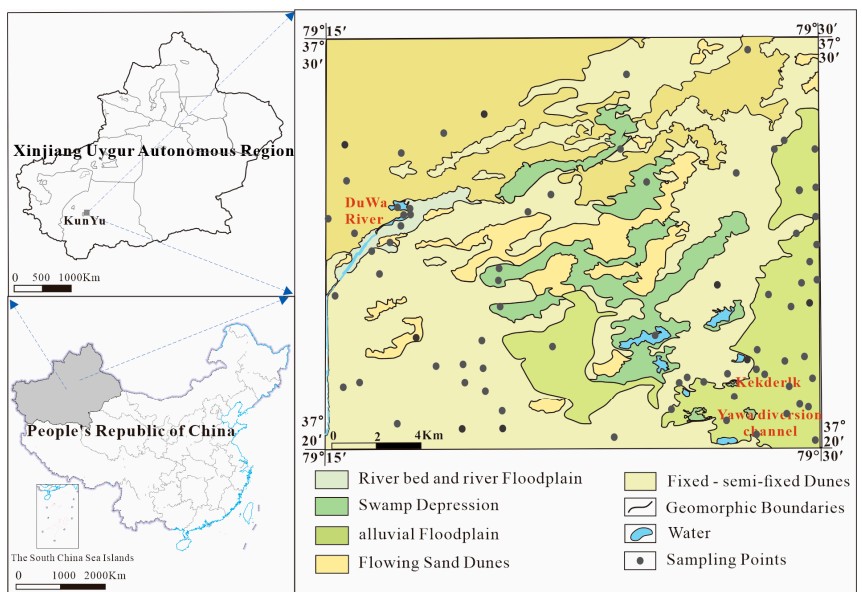

**Figure 1.** Geomorphology type and location of sampling sites in the study area.

According to field geological surveys and remote sensing image data, the land utilization rate in the research area is relatively low. The southeast and southwest of the study area are mainly residential areas, the southern and central eastern parts are mainly farmland, with jujube trees planted, and the northern parts are vast deserts that have not been utilized.

### 2.2. Hydrogeological Conditions

According to the lithology and hydrogeological characteristics of Quaternary sediments, combined with pumping tests and previous hydrological borehole data, the Quaternary Holocene (Qh)-Pleistocene (Qp) aquifer group in the study area belongs to pore water in loose rocks. The aquifer is primarily composed of fine sand and silt, with a thickness ranging from 50 to 76 m (the comprehensive histogram in the study area is shown in

Figure S1 of the Supplementary Materials). The depth of the water table varies significantly across different geomorphic types. The flowing dune area in the northwest has a depth of over 5 m, while the fixed and semi-fixed dune area in the southwest and northeast ranges from 3 to 5 m. In the alluvial plain, the water table depth is 1–3 m, whereas, in the central swampy depressions and some parts of the northwest riverbed and floodplain, the depth is less than 1 m. Generally, the water table depth decreases from northwest to southeast, as shown in Figure 2. Combined with the hydrogeological borehole and previous geological data, the area shows a distribution pattern of weak water richness in the northwest and medium in the southeast.

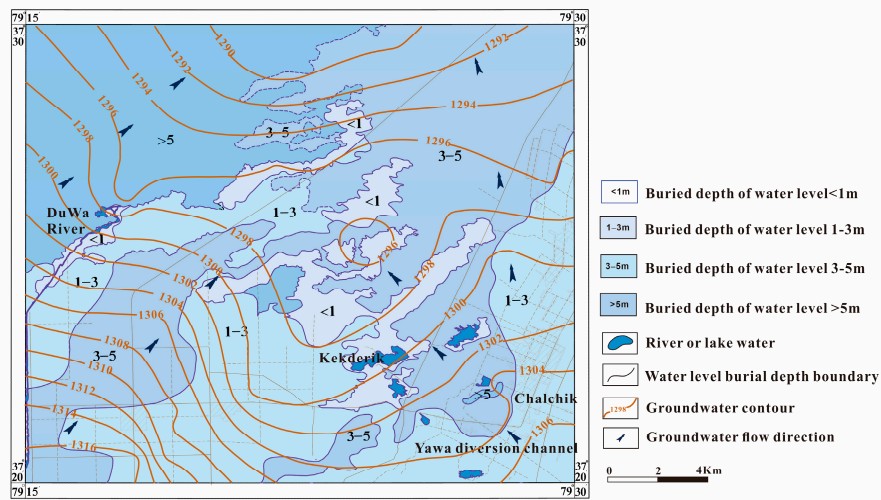

**Figure 2.** Contour map of phreatic water depth and water level.

In terms of recharge sources, the phreatic water in the area is mainly recharged laterally from the Duwa River, as well as from irrigation infiltration and groundwater lateral recharge. The main discharge methods include artificial mining, evaporation, and transpiration. The overall flow direction is from southwest to northeast, but local areas are affected by rivers, canal systems, field irrigation and groundwater exploitation, which makes the groundwater flow field change to some extent. The ground slope of the floodplain area outside the southern part of the area is small, about 1.67%, and the aquifer particles are mainly sand gravel and gravelly silt. After entering the working area to the north, the slope of the terrain continues to decrease, to only 0.15~0.30‰. Furthermore, the lithology of the aquifer becomes fine sand, and the groundwater flow rate gradually becomes smaller. In Yawa Township in the eastern part of the study area, diving flows from the southeast to the northwest and north under the influence of the Krakash River and the Yawa Dry Drain. The burial depth of groundwater mostly ranges from 1 to 3 m, with a slight topographic slope, and the lithology of the aquifer consisting of fine powder sand, and a hydraulic slope of 0.5‰ to 2‰. The burial depth of groundwater in the middle is mostly less than 1 m, and the groundwater runoff rate is sluggish. In the vertical direction, the shallow vertical runoff is stronger. With the increase in depth, the aquifer compactness increases and the groundwater runoff weakens.

### 2.3. Sample Collection and Testing

The specific sampling location is shown in Figure 1. Ninety (90) groups of groundwater samples were analyzed altogether in this study, which were obtained from machinery wells, hydrological boreholes, and the Duval River in the study area. The sample collection time was concentrated in late July 2021, with two water samples taken from each borehole and only one sample taken from other locations. The water samples were collected in 500 mL narrow-mouth polyethylene bottles. Before sampling, the sample bottles were washed with the water to be tested 2–3 times. After sampling, the sample bottles were sealed to

prevent leakage and sent to the laboratory for testing as soon as possible. The water sample testing was completed by the Experimental Testing Center of Xi'an Geological Survey Center of China Geological Survey. The main testing indexes included pH, total dissolved solids (TDS), $K^+$, $Na^+$, $Ca^{2+}$, $Mg^{2+}$, $Cl^-$, $SO_4^{2-}$, $HCO_3^-$, $NO_3^-$, $F^-$, total hardness (TH), chemical oxygen demand (COD) and other related components. Electrical conductivity (EC), total dissolved solids (TDS), and pH of the water samples were measured in situ using a pH meter and portable EC instrument. In the laboratory, all water samples were filtered through 0.45 μm membrane filters to separate suspended particles before testing. Major anions ($F^-$, $Cl^-$, $SO_4^{2-}$, $NO_3^-$) were tested using ion chromatography; major cations ($Ca^{2+}$, $Mg^{2+}$, $Na^+$, $K^+$) were tested by inductively coupled plasma mass spectrometry, and bicarbonate ($HCO_3^-$) concentrations were measured using an automatic titrator. TH were measured using EDTA complexometric method. The COD value was determined by potassium permanganate method. The accuracy of test results was examined according to the charge balance error $E$ [27], and it was maintained within 5% for all.

### 2.4. Analysis Method

IBM SPSS 23.0 software was used for the statistical analysis of the data. The groundwater was classified using the Shukarev classification method, and the map of groundwater mineralization distribution and hydrochemical types area was drawn using MapGIS software based on kriging interpolation. The primary ion source of groundwater and the influencing factors of hydrochemical characteristics were determined through the Gibbs diagram and major ion ratio relationship. The quality of drinking water was evaluated by the WQI, and the quality of irrigation water was evaluated by sodium percentage (Na%), sodium adsorption ratio (SAR), permeability index (PI) and magnesium hazard (MH).

### 2.4.1. Water Quality Index Method (WQI)

WQI can express the comprehensive impact of different water quality parameters on the suitability of drinking water. These parameters were given different weight values ($w_i$) according to the impact on water quality. Through the analysis of sample test results, nine of these indicators were selected as evaluation factors, and their weights were distributed as follows: TDS and $NO_3^-$ were 5, pH, $SO_4^{2-}$, COD and $F^-$ were 4, $Cl^-$ was 3, $Na^+$ was 2, and total hardness (TH) was 1 [28–30].

The water quality index (WQI) can be determined by calculating the relative weight ($W_i$), quality rating ($q_i$) and sub-index ($SI_i$) of each parameter using the equation.

$$W_i = \frac{w_i}{\sum_{i=1}^{n} w_i} \tag{1}$$

$$q_i = \frac{C_i}{S_i} \times 100 \tag{2}$$

$$SI_i = W_i \times q_i \tag{3}$$

where: $W_i$ is the relative weight, $w_i$ is the weight of each parameter, and $n$ is the number of parameters; $Q_i$ is the quality rating, $C_i$ is the value or concentration of each parameter in each water sample; $S_i$ is the drinking water standard of each parameter, and $SI_i$ is the sub-index of each parameter.

WQI is the sum of all sub-indexes $SI_i$ calculated for each parameter of the sample:

$$\text{WQI} = \sum_{i=1}^{n} SI_i \tag{4}$$

### 2.4.2. Evaluation Method for Suitability of Irrigation Water

When evaluating the suitability of irrigation water, salinity, alkalinity and toxicity are usually considered. High salinity irrigation water will bring about the emergence of saline soil, while high alkalinity irrigation water will cause soil alkalization and hardening, and

high sodium will increase soil alkalinity. The sodium hazard of irrigation water can be expressed by the sodium adsorption ratio (SAR). A higher SAR value indicates that the alkalinity of irrigation water is high [13]. Similarly, the percentage of sodium (Na%) is also a standard index for evaluating the suitability of irrigation water and can be used as the basis for the classification of irrigation water.

If the concentration of $Mg^{2+}$ in irrigation water reaches a high level, magnesium alkalization may occur, thus affecting the organic matter content in the soil and causing crop yield reduction. Therefore, Szaboles and Darab proposed magnesium hazard (MH) to evaluate the applicability of agricultural irrigation water [31]. The permeability index (PI) is another important parameter to measure the suitability of groundwater for irrigation. It represents the ability of water and minerals to migrate through pore space to the plant roots, and is mainly influenced by dissolved ions such as $Na^+$, $Ca^{2+}$, $Mg^{2+}$ and $HCO_3^-$, based on which Doneen developed the relevant evaluation criteria according to PI [32]. The calculation formulas of each indicator are listed in Table 1.

**Table 1.** Evaluation index and calculation formula of irrigation water suitability.

| Evaluation Parameters | Calculation Formulas |
| --- | --- |
| SAR | $SAR = Na/[(Ca + Mg)/2]^{0.5}$ |
| Na% | $Na\% = (Na + K)/(Ca + Mg + Na + K) \times 100\%$ |
| MH | $MH = Mg/(Ca + Mg) \times 100\%$ |
| PI | $PI = (Na + \sqrt{HCO_3})/(Ca + Mg + Na) \times 100\%$ |

## 3. Results and Discussion

### 3.1. Basic Characteristics of Hydrochemistry

The physicochemical parameters of groundwater in the study area, including statistical indicators such as minimum, maximum, mean and standard deviation, are given in Table 2. From the overall content of each ion, the cation order is $Na^+ > Mg^{2+} > Ca^{2+} > K^+$, of which the content of $Na^+$ is the largest, with a mean concentration of 696.1 mg/L, which is 5.2, 5.8 and 19.0 times of the content of $Mg^{2+}$, $Ca^{2+}$ and $K^+$, respectively. The anion content is ranked as $Cl^- > SO_4^{2-} > HCO_3^- > NO_3^- > F^-$, of which $Cl^-$ and $SO_4^{2-}$ are the primary anions in groundwater, with an average content of 950.4 mg/L and 824.1 mg/L, respectively. The pH of water samples ranges from 6.9 to 9.3, with a mean value of 7.9, which is weakly alkaline overall, and its coefficient of variation (*CV*) is only 4.9%, indicating that the spatial variability of groundwater pH is small. The total hardness (TH) of the water samples varies between 126.4 and 35,350.0 mg/L, with an average value of 1232.8 mg/L. The overall hardness of the groundwater is relatively high, dominated by hard water.

**Table 2.** Statistical results of main hydrochemical parameters of groundwater.

| Project | $Na^+$ | $K^+$ | $Ca^{2+}$ | $Mg^{2+}$ | $Cl^-$ | $SO_4^{2-}$ | $HCO_3^-$ | $NO_3^-$ | $COD_{Mn}$ | $F^-$ | TDS | TH | pH | EC |
| --- | --- | --- | --- | --- | --- | --- | --- | --- | --- | --- | --- | --- | --- | --- |
| Min | 54.5 | 1.8 | 10.9 | 10.9 | 49.4 | 71.3 | 51.7 | 0.1 | 0.8 | 0.3 | 714 | 126.4 | 6.9 | 1320.0 |
| Max | 4830.0 | 171.0 | 592.0 | 750 | 7575.0 | 6416.0 | 913.0 | 354.0 | 7.0 | 2.9 | 19,100.0 | 35,350.0 | 9.3 | 8660.0 |
| Mean | 696.1 | 36.8 | 118.3 | 135.0 | 950.4 | 824.1 | 341.9 | 15.2 | 2.3 | 1.2 | 3028.7 | 1232.8 | 7.9 | 3953.4 |
| SD | 821.4 | 30.7 | 93.6 | 103.0 | 1139.6 | 872.4 | 212.3 | 45.3 | 1.6 | 0.9 | 3030.3 | 3665.5 | 0.4 | 1842.9 |
| *CV%* | 118.0 | 83.5 | 79.1 | 76.3 | 119.9 | 105.9 | 62.1 | 297.3 | 68.1 | 74.5 | 100.0 | 297.3 | 4.9 | 46.6 |

Note: pH has no unit, the unit of EC is μs/cm, and the unit of other parameters is mg/L.

The distribution map of TDS in the research area is shown in Figure 3. The variation of TDS ranges from 714.0 to 19,100.0 mg/L, and the average value is 3028.7 mg/L. According to its classification standard [33], the groundwater in study area mainly belongs to brackish water and semi-saline, and its distribution area accounts for 63.06% and 20.06% of the total area, respectively. Freshwater is scattered in the southeastern of the study area, making up only 2.59% of the total area. In comparison, saline water is mainly distributed near the Kekderik, with a strip distribution area of about 82.15 km$^2$, accounting for 20.06%. Mainly due to the slow groundwater runoff, shallow water level, and strong evaporation, the TDS

value in this area is relatively high. COD is an indicator that characterizes organic pollution in water. The $COD_{Mn}$ range in the study area is between 0.8 and 7.0, with an average of 2.3. Most water samples have low CODMn values, which comply with the Class III water regulations in the Sanitary Standard for Drinking Water (GB5749-2006) [34]. Only seven water samples exceed the Class III water limit.

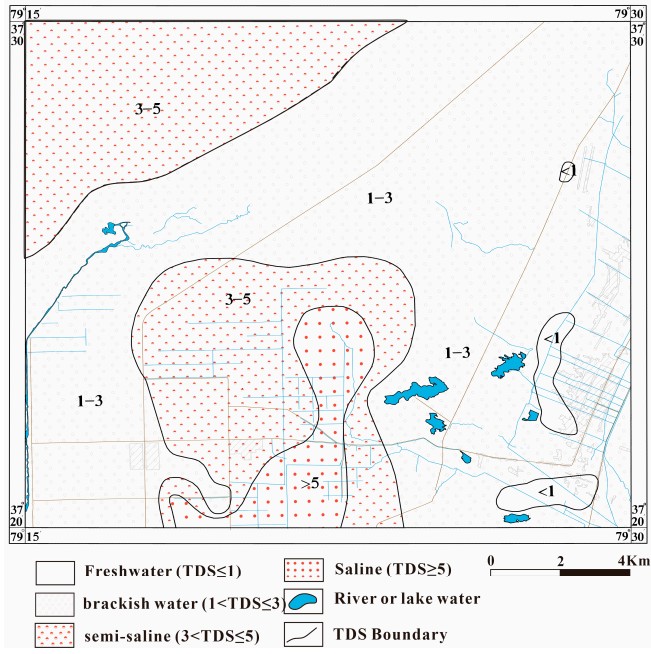

**Figure 3.** Distribution map of groundwater TDS value.

Correlation analysis can show the close degree between different hydrochemical parameters, which is helpful to better understand the hydrochemical process and ion source. The Pearson correlation analysis results of various indicators are shown in Figure 4. It can be observed from the figure that $Na^+$ has a strong correlation with $K^+$, $Mg^{2+}$, $SO_4^{2-}$, $Cl^-$ and TDS, indicating that sodium ions may have many sources. $Ca^{2+}$, $Mg^{2+}$ strongly correlate with $SO_4^{2-}$, but have a poor correlation with $HCO_3^-$. TDS has a strong positive correlation with $Cl^-$, indicating that saline rock may be dissolved along the flow path [35]. However, the correlation between $NO_3^-$ and other parameters is weak because its source is mainly artificial, such as the application of nitrogen fertilizer in jujube trees.

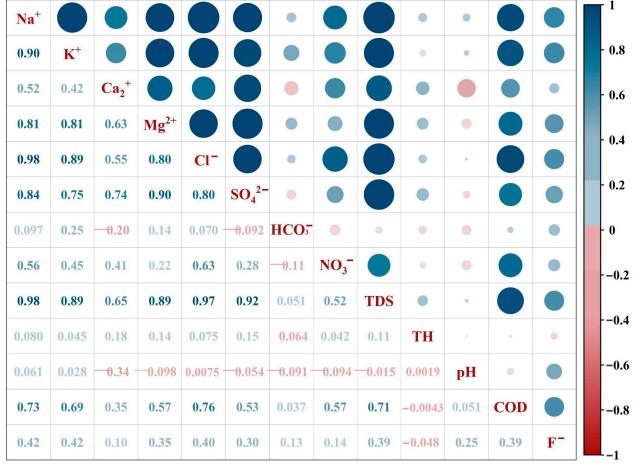

**Figure 4.** Correlation matrices between parameters.

*3.2. Hydrochemical Evolution Characteristics*

The Shukarev method is used to classify hydrochemical types based on the concentration and mineralization degree of six common major ions in groundwater. According to the Shukarev method classification [36], the hydrochemical types in the study area can be divided into seven categories (Figure 5). The Cl-SO$_4$-Na-Mg type and Cl-Na type are predominant, among which the Cl-SO$_4$-Na-Mg type groundwater is generally distributed in the flowing dunes and fixed-semi-fixed dunes. The groundwater runoff conditions in these areas are poor, and water alternation is medium. The concentration of Cl$^-$, SO$_4^{2-}$, Na$^+$, and Mg$^{2+}$ in groundwater is relatively high, while the content of other ions is relatively low. The Cl-Na type groundwater is found in the northeast desert and the vicinity of Kekderik. Its groundwater depth is shallow, runoff is slow, evaporation is strong, local groundwater is stagnant, salt accumulation is serious, and Na$^+$ content is high. The groundwater in the southern swamp depression primarily belongs to the Cl-SO$_4$-Na type. Due to the influence of Karakash River and Yawa diversion channel, the groundwater in the eastern Yawa Township is mainly of Cl-HCO$_3$-Na-Mg type and Cl-HCO$_3$-SO$_4$-Na-Mg type. In addition, Cl-Na-Mg type groundwater is only sporadically distributed in the east near Chalchik.

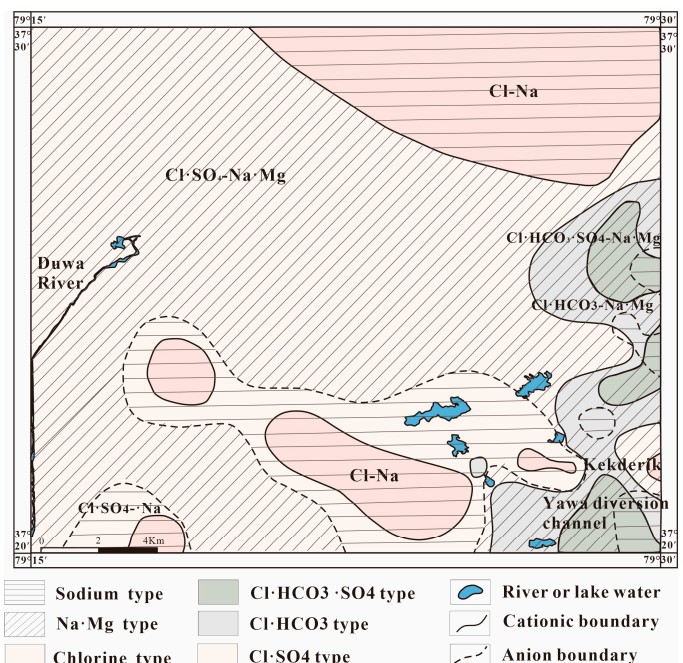

**Figure 5.** Hydrochemical types of groundwater in the study area.

Figure 6 shows the piper trilinear diagram of groundwater hydrochemistry in the study area. In this diagram, water samples are relatively concentrated, basically falling into Zones IV and II, and more than 90% are distributed in Zone IV. The main anions are close to the Cl$^-$ and SO$_4^{2-}$ ends, and their milligram equivalent percentages are mostly between 40% and 70%, with strong acids (Cl$^-$, SO$_4^{2-}$) exceeding weak acids (HCO$_3^-$). The main cations are close to the Na$^+$ and Mg$^{2+}$ ends, followed by Ca$^{2+}$, and the concentration of alkali metal ions (Na$^+$, K$^+$) is higher than that of alkaline earth metal ions (Ca$^{2+}$, Mg$^{2+}$). The hydrochemical types mainly showed Cl-SO$_4$-Na type and Cl-Na type. This result is similar to the water chemistry types classified by the Shukarev method.

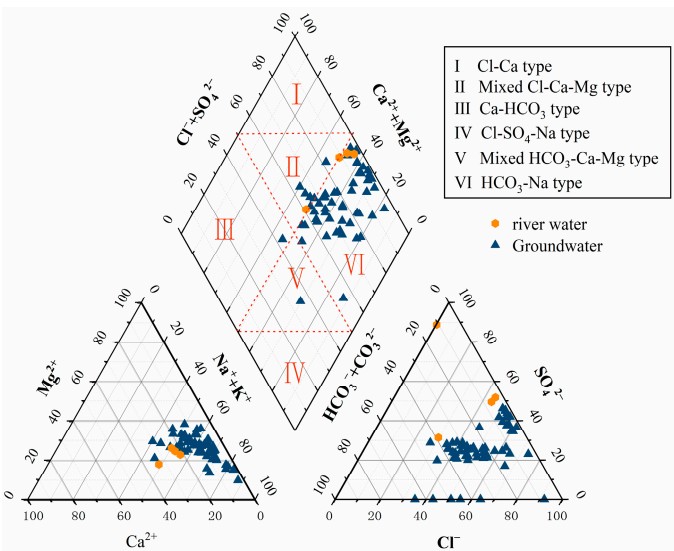

**Figure 6.** Piper diagram of groundwater chemical composition in the study area.

### 3.3. Analysis of Hydrochemical Origin of Groundwater

The influence of rock weathering, atmospheric precipitation, evaporation and concentration on groundwater hydrochemistry can be qualitatively analyzed by the Gibbs diagram [37]. In the Gibbs diagram, when the TDS value is low and the ratio of $\gamma Na^+/\gamma(Na^+ + Ca^{2+})$ or $\gamma Cl^-/\gamma(Cl^- + HCO_3^-)$ is greater than 0.5, the groundwater chemical components are mainly influenced by atmospheric precipitation [38]. While higher TDS values and also high $\gamma Na^+/\gamma(Na^+ + Ca^{2+})$ ratio or $\gamma Cl^-/\gamma(Cl^- + HCO_3^-)$ ratio indicate that the groundwater chemical fraction is affected by evaporative concentration, otherwise by rock weathering. It can be seen from Figure 7 that the TDS values of water samples in the study area are high and 98% of them fall within the range of $\gamma Na^+/\gamma(Na^+ + Ca^{2+}) > 0.5$, while the ratio of $\gamma Cl^-/\gamma(Cl^- + HCO_3^-)$ is relatively dispersed, and parts of samples distributed in the region of rock weathering. This shows that the groundwater in this area is affected by both evaporation concentration and rock weathering, and the former is more significant.

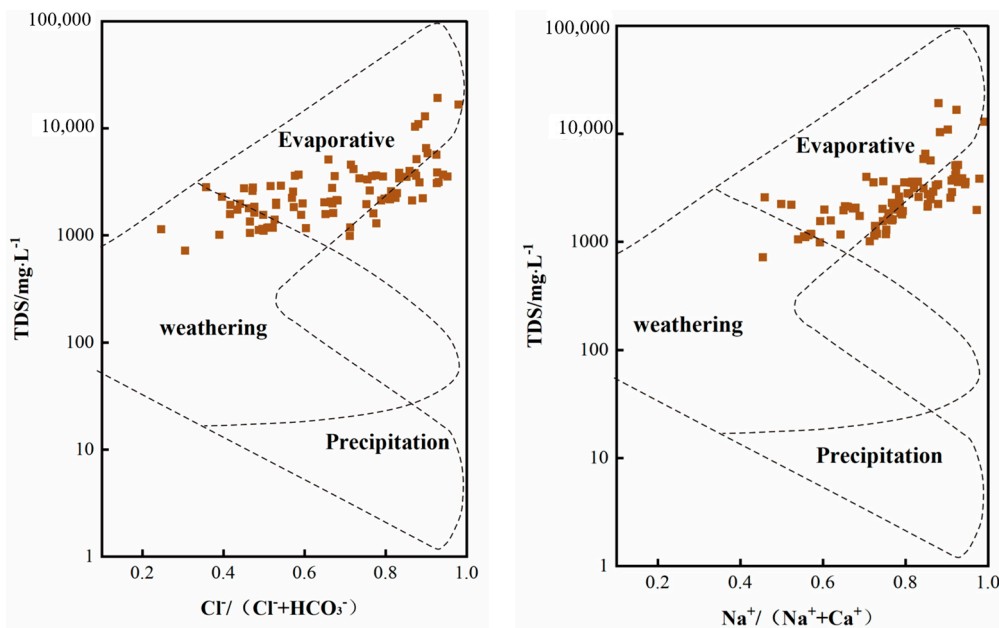

**Figure 7.** Gibbs diagram of groundwater in the study area.

Groundwater formed under different genesis or conditions often has a large difference in the content ratio between components. Therefore, such ratio coefficients can be used to determine the source of groundwater components. Among them, the relationship of $\gamma(Na^+ + K^+)/\gamma Cl^-$ can be used to determine the sources of $K^+$ and $Na^+$ in groundwater. When the $\gamma(Na^+ + K^+)/\gamma Cl^-$ distribution is around the line y = x, the $K^+$ and $Na^+$ in groundwater are primarily from rock salt dissolution [39]. Figure 8a shows that $K^+$ and $Na^+$ in groundwater in the study area mainly originate from rock salt dissolution, 90% of the water samples fall in the zone where $\gamma(Na^+ + K^+)/\gamma Cl^- > 1$, only a few water sample points fall below. This indicates that the content of $Na^+$ and $K^+$ is more than that of $Cl^-$, which may be caused by the dissolution of silica-aluminate minerals containing sodium and potassium (e.g., feldspar), or there may be alternating cation adsorption.

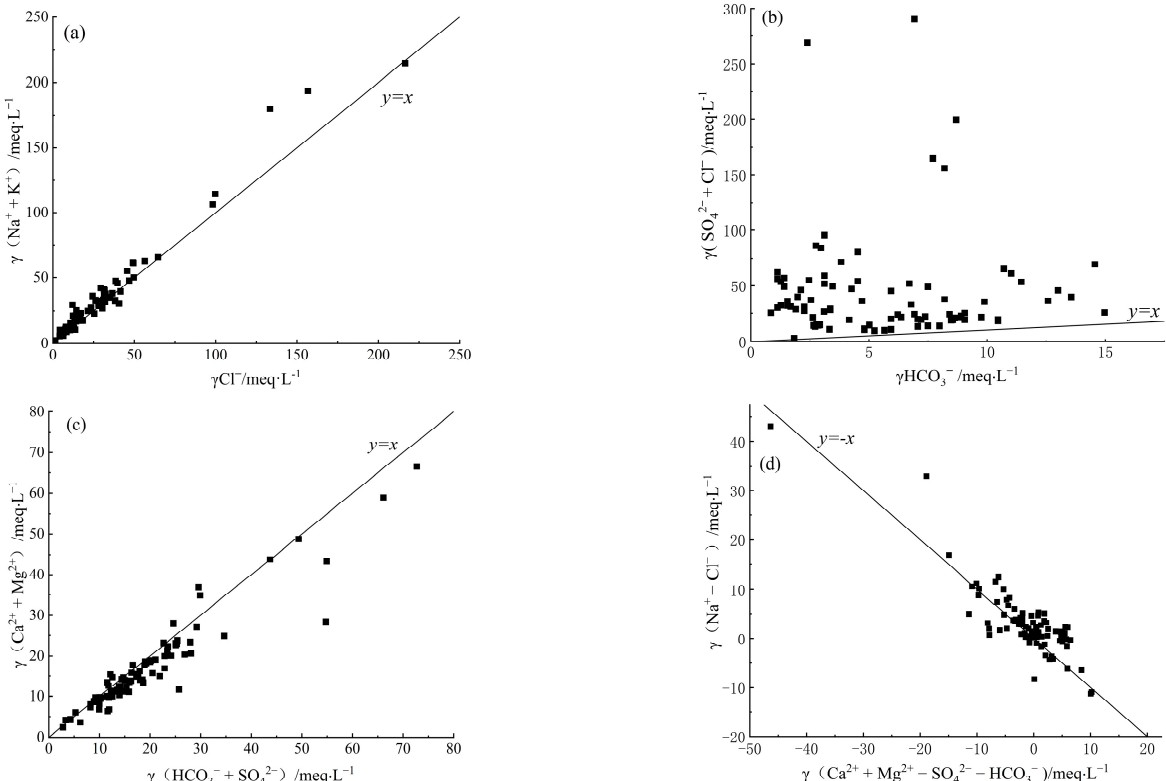

**Figure 8.** Groundwater ion ratio diagram. (**a**) Scatter plot of $\gamma(Na^+ + K^+)$ versus $\gamma Cl^-$; (**b**) Scatter plot of $\gamma(SO_4^{2-} + Cl^-)$ versus $\gamma HCO_3^-$; (**c**) Scatter plot of $\gamma(Ca^{2+} + Mg^{2+})$ versus $\gamma(HCO_3^- + SO_4^{2-})$; (**d**) Scatter plot of $\gamma(Na^+ - Cl^-)$ versus $\gamma(Ca^{2+} + Mg^{2+} - HCO_3^- + SO_4^{2-})$.

The dissolution of carbonate rocks in groundwater can be judged by the relationship of $\gamma(SO_4^{2-} + Cl^-)/\gamma(HCO_3^-)$ [39]. As seen from Figure 8b, the groundwater samples all fall above the line y = x, indicating that the chemical composition of groundwater is mainly influenced by the dissolution of evaporite, and the influence of carbonate rocks is weak. The main sources of $Mg^{2+}$ and $Ca^{2+}$ can be determined by analyzing the ratio relationship between $\gamma(Ca^{2+} + Mg^{2+})$ and $\gamma(HCO_3^- + SO_4^{2-})$ [40]. Figure 8c indicates that over 80% of the groundwater samples are located below $\gamma(Ca^{2+} + Mg^{2+})/\gamma(HCO_3^- + SO_4^{2-})$, indicating that $Mg^{2+}$ and $Ca^{2+}$ in groundwater of the study area are primarily from the weathering dissolution of silicate and evaporite, and only a tiny part is related to the weathering dissolution of carbonate rocks. Meanwhile, in Figure 4, both $Ca^{2+}$ and $Mg^{2+}$ are strongly correlated with $SO_4^{2-}$, which further explains that the dissolution of gypsum and magnesium sulfate is an important source of $Ca^{2+}$ and $Mg^{2+}$.

The ratio relationship between $\gamma(Na^+ - Cl^-)$ and $\gamma(Ca^{2+} + Mg^{2+} - SO_4^{2-} - HCO_3^-)$ is usually used to reveal the cation exchange rate [40]. Groundwater samples fall near the line y = −x, confirming the existence of cation exchange adsorption. The strength

and direction of cation exchange can be inferred from the CAI-1 and CAI-2 indices in the Scholler method [41], which are calculated by Equations (5) and (6), respectively:

$$CAI\text{-}1 = \frac{C(Cl^-) - [C(Na^+) + C(K^+)]}{C(Cl^-)} \tag{5}$$

$$CAI\text{-}2 = \frac{C(Cl^-) - [C(Na^+) + C(K^+)]}{C(HCO_3{}^-) + C(SO_4{}^{2-}) + C(CO_3{}^-) + C(NO_3{}^-)} \tag{6}$$

where: C is the ion concentration, in meq/L.

When CAI-1 and CAI-2 are all positive values, it suggests that there is a positive cation exchange, that is, $Na^+$ and $K^+$ in water exchange with $Ca^{2+}$ and $Mg^{2+}$ on the surface of mineral particles. Otherwise, if values are negative, the reverse cation exchange occurs, and the greater the absolute value, the stronger the cation alternation [42]. CAI-1 and CAI-2 are negative, with average values of $-0.52$ and $-0.47$, respectively. The absolute value is relatively small, indicating that reverse cation alternation has occurred, and this effect is relatively weak.

The analysis results of the ion combination ratio show that $Na^+$ and $K^+$ mainly come from salt rock and silica-aluminate minerals dissolution, $Mg^{2+}$ and $Ca^{2+}$ mainly come from weathering dissolution of silicate rocks and evaporite, and $SO_4{}^{2-}$ mainly comes from sulfate dissolution, which is more consistent with the regional geological background. In the south of the study area, the Late Permian K-feldspar granite strata are exposed near Kunlun Mountain, and the Jurassic, Cretaceous and Paleoproterozoic strata in the area contain evaporites such as gypsum and rock salt of varying thickness. At the same time, a large-scale gypsum mine is developed in the Piaman anticline near the south of Kunyu city. Carbonate rocks have a feeble influence on the chemical composition of groundwater, and it is assumed that desulfurization is one of the important sources of $HCO_3{}^-$ in water.

### 3.4. Drinking Water Quality Evaluation

The drinking water quality in the study area was comprehensively evaluated with the Sanitary Standard for Drinking Water (GB5749-2006) [34] as an evaluation index. The limit values and weight values of each factor are shown in Table 3. Meanwhile, the WQI spatial distribution map was drawn according to the calculation results (Figure 9).

Based on the WQI values, groundwater quality can be classified as excellent water (<50), good water (50–100), poor water (100–200), very poor water (200–300) and unsuitable (>300) [43]. The WQI values in the study area range from 43.0 to 1442.9, with a mean value of 224.2, which is relatively large. Only 16 samples belong to the grade of excellent water to good water, and 11 samples belong to the unsuitable category. The rest of the water samples are poor or very poor water. The WQI spatial distribution map shows that most areas in the study area have high WQI values and are not suitable for drinking. Only the groundwater in parts of the eastern regions has low WQI values and is suitable for drinking.

**Table 3.** List of parameters, weight factors and limits of water quality index.

| Indicators | TDS | $NO_3{}^-$ | $SO_4{}^{2-}$ | COD | PH | $F^-$ | $Cl^-$ | $Na^+$ | TH |
|---|---|---|---|---|---|---|---|---|---|
| Limit value | 1000 mg/L | 10 mg/L | 250 mg/L | 3 mg/L | 6.5–8.5 | 1 mg/L | 250 mg/L | 200 mg/L | 450 mg/L |
| Weight | 5 | 5 | 4 | 4 | 4 | 4 | 3 | 2 | 1 |
| Relative weights | 0.156 | 0.156 | 0.125 | 0.125 | 0.125 | 0.125 | 0.094 | 0.063 | 0.031 |

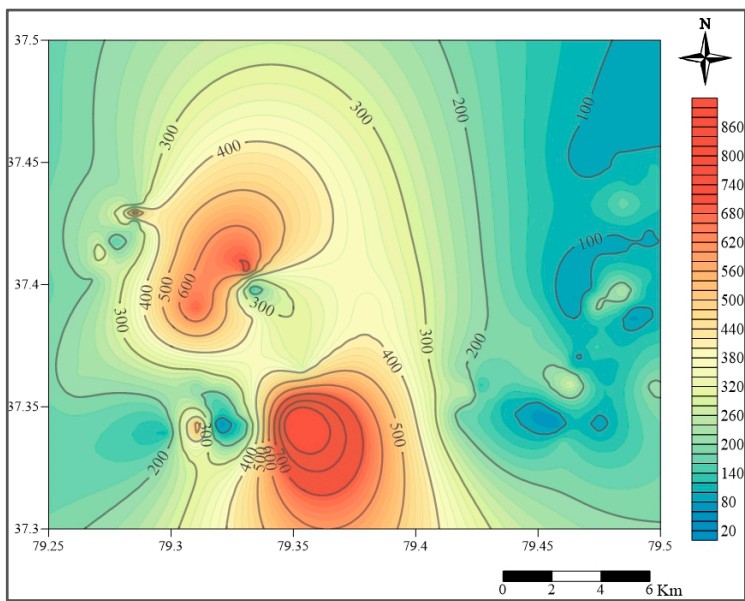

**Figure 9.** Spatial distribution of water quality index (WQI).

The spatial distribution map of WQI shows that the water quality in the central part of the study area is in the unsuitable category, and two high value areas occur in the lower DuWa River area and near Kekderik. The slow groundwater runoff, shallow depth of burial and strong evaporation in the vicinity of Kekderik result in high TDS values, and the TDS values have a greater weight on the WQI values, which lead to the high WQI values in this area. The sampling survey along the river from the upstream to the downstream of the Duval River found that the water near the starting survey site belonged to brackish. However, the TDS value along the way gradually increased, and the TDS content reached 3960 mg/L at the tail, which was semi-saline water. It reflects that the ion concentration of Duwa River increases with the runoff distance, among which $SO_4^{2-}$, $Na^+$ content and TDS value increase more significantly. The river water with high ion concentration recharges the groundwater, causing an increase in the corresponding ion concentration of groundwater in the downstream area; in addition, the dissolution of gypsum mine near this area also has a certain influence on the groundwater quality. The eastern part of the study area is scattered with excellent and good water areas. This is mainly due to the swampy depressions in the area, which mainly receive recharge from irrigation infiltration and river channel infiltration, and the main mining method is artificial mining. The groundwater alternation is stronger, the influence of evaporation is relatively weak, and the ion concentration value is relatively low.

### 3.5. Irrigation Water Quality Evaluation

Salinity has a significant effect on soil alkalinity and crop growth, and when irrigated with high salinity groundwater, it can lead to soil hardening and inhibit crop growth [44]. The hazard of salinity and alkalinity can be better understood using the United States Salinity Laboratory diagrams (USSL diagrams), where salinity and alkalinity are expressed in terms of electrical conductivity (EC) and sodium adsorption ratio (SAR), respectively [13]. The USSL classification map of the study area was plotted (Figure 10), in which 27% of the groundwater samples distributed in the high salinity region and the remaining 73% fell in the very high salinity region. The SAR values varied between 1.64 and 30.33, with a mean value of 8.47, which were evenly dispersed in the four classification regions S1–S4, and about half of the water samples fell in the high-very high alkalinity region. In summary, the study found that the groundwater in the study area has high-very high salinity overall, but the alkalinity is highly variable and the SAR value is high in some areas, which requires certain measures before it can be used for irrigation.

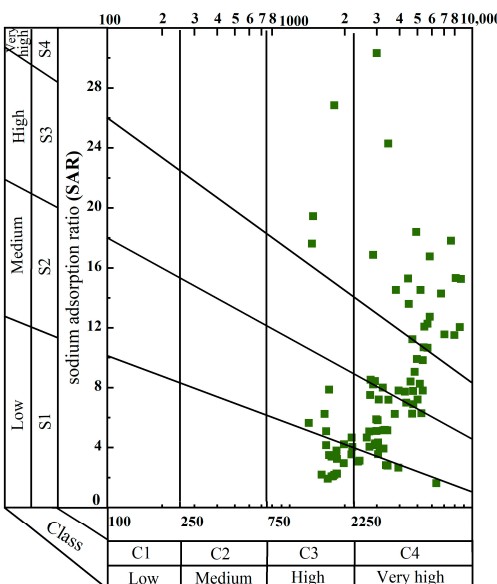

**Figure 10.** USSL classification chart of groundwater irrigation water quality.

Wilcox plots reflecting salinity and sodium percentage (Na%) allow for the classification of irrigation water [45]. Plotting the samples data into the Wilcox plot (Figure 11), it is found that about 56% of the groundwater samples are in the unsuitable zone, 20% are in the doubtful-unsuitable category, 6% are in the permissible-doubtful category, and only 18% are in the good-permissible category. Most water samples fall within the doubtful or unsuitable range, indicating that the groundwater in this area is not suitable for direct irrigation, mainly due to the high salinity of the groundwater.

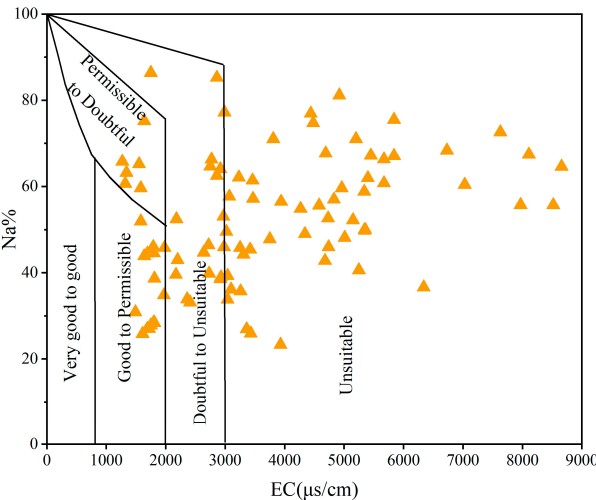

**Figure 11.** Suitability of groundwater irrigation based on Na%.

Soil permeability is affected by long-term water use and the concentration of calcium, magnesium, sodium, and bicarbonate in the water. The Doneen diagrams based on total salt concentration and permeability index (PI) can be used to characterize the suitability of groundwater for irrigation. In this regard, the PI can be classified into three categories. Class I and II have a maximum permeability of 75% or more and are suitable for irrigation, while class III water indicates a maximum permeability of 25% of the soil and is unsuitable for irrigation [46]. As shown in Figure 12, most of the groundwater samples in the study area belong to category I and only five samples belong to category II, indicating that from the irrigation index (PI), long-term irrigation of groundwater has little impact on soil permeability.

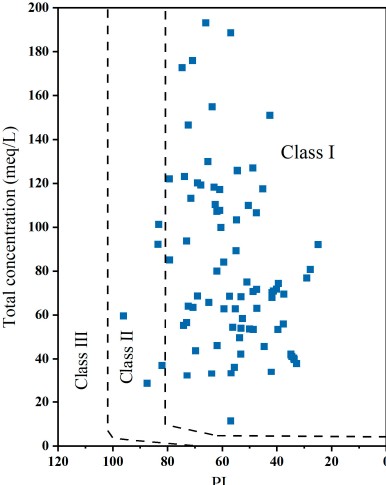

**Figure 12.** Doneen diagram of groundwater irrigation water quality.

The concentration of magnesium in water is one of the most important parameters to test the suitability of irrigation water. Calcium and magnesium must be present in suitable concentrations to maintain a balance between them. The increase in magnesium concentration in irrigation water will lead to an increase in alkalinity and a decrease in crop yield. Therefore, water with MH value greater than 50% is considered unsuitable for irrigation. The calculated MH values in the study area range from 21.0% to 90.2%, with a mean value of 50.6%, of which 54% of the samples were above the MH limit, indicating that the groundwater magnesium concentration was high and unsuitable for irrigation.

## 4. Conclusions

In this study, mathematical statistics, Shukalev classification, Gibbs model and ion ratio were applied to analyze the ion characteristics and evolution mechanism of shallow groundwater in Kunyu desert area in the southern edge of Tarim Basin. The research results indicate that the abundance of major cations was in the order of $Na^+ > Mg^{2+} > Ca^{2+} > K^+$, while the major anions trend in the study area was in the following order: $Cl^- > SO_4^{2-} > HCO_3^- > NO_3^- > F^-$. Influenced by the depth of the water table and strong evaporation, the TDS value of groundwater is relatively high and is primarily semi-saline water. The hydrochemical type in the study area is mainly the $Cl-SO_4-Na-Mg$ type. The ions in groundwater are mainly derived from the weathering dissolution of saline rock, silicates and sulfates. The hydrochemical characteristics are mainly controlled by the combination of evaporation concentration and rock weathering. In addition, the desulfation may also have some influence on it.

A comprehensive evaluation of drinking water quality was conducted based on WQI. The overall WQI value in the study area is relatively high, and only parts of the eastern regions have groundwater suitable for drinking. Two high-value WQI areas occur in the lower DuWa River and near Kekdric, which is residential area and farmland. Residents in this area should not use untreated groundwater as drinking water.

Through the USSL classification chart, Wilcox chart, Doneen chart and combining with the SAR and other indicators to evaluate the water quality of irrigation water, it is found that although the PI of groundwater in the study area is suitable, the overall salinity is high-very high, the alkalinity is greatly variable, and the magnesium concentration is high. Therefore, the direct use of groundwater for irrigation will lead to soil hardening and low crop yield, and certain measures need to be taken before it can be used for irrigation.

These water samples are from the same batch, and the sampling time is concentrated in late July 2021. Due to the difficulty of drilling and sampling deep into the desert, there are few samples taken in the northwest region of the study area. Based on data from other parts of the study area, the water level depth values inferred using interpolation methods may slightly differ from the actual values. Furthermore, in a future project research,



batch sampling in different seasons will be considered, so as to compare and study the hydrochemical characteristics and water quality status under different seasons and climatic conditions.

**Supplementary Materials:** The following supporting information can be downloaded at: https://www.mdpi.com/article/10.3390/w15081563/s1, Figure S1: Comprehensive histogram in the study area.

**Author Contributions:** All authors contributed to the study's conception and design. S.D. provided the writing ideas and supervised the study; M.Z. collected water samples and sent them for inspection; Z.Z., M.B., C.Z. and P.L. processed and analyzed the data. The first draft of the manuscript was written by R.T., and all authors commented on previous versions of the manuscript. All authors have read and agreed to the published version of the manuscript.

**Funding:** This research is supported by the National Natural Science Foundation of China (Grant No. 41807221), Tiandi Science and Technology Co. Ltd. Science and Technology Innovation Venture Capital Special Project (2022-2-TD-ZD005), Project of China Geological Survey (DD20179605).

**Data Availability Statement:** Not applicable.

**Acknowledgments:** The authors are grateful to the anonymous reviewers for their helpful comments on the manuscript.

**Conflicts of Interest:** The authors declare no conflict of interest.

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
