# Peer review of "Hydrochemical Characteristics and Water Quality of Shallow Groundwater in Desert Area of Kunyu City, Southern Margin of Tarim Basin, China"

_water, doi:10.3390/w15081563_

Round 1
Reviewer 1 Report
--- The conclusion part should be rewritten. It was not very pleasant to be in substance.
--- Add the limitations of the study and your suggestions in conclusion.
--- A discussion section should be written by citing studies in the literature. Identify the differences of this study by referring to other studies in your country.
--- Add the coefficient of determination of the regression equation to Figure 8.
--- Indicate the location of the study area in the world.
--- The introduction is very weak. There are only 15 articles in the introduction. Replicate the number of articles to cover the last 5 years.
--- What is the purpose of this study?
Reviewer 2 Report
Manuscript ID: water-2300204
Title: Quality and Hydrochemical Characteristics of Shallow Groundwater in Desert Area of Kunyu City, Southern Margin of Tarim Basin
by Runchi Tang, Shuning Dong, Mengfei Zhang, Zhenfang Zhou, Chenghang Zhang, Pei Li and Mengtong Bai
Written evaluation
Overall impression and recommendation
The manuscript investigates hydrochemical characteristics and groundwater quality in desert area. The paper is generally well structured, but I have some concerns, accompanied with comments which need to be addressed as stated in general and specific comments bellow.
General comments
1) language editing required
I noticed some grammatical errors in the paper, such as the word order, use of articles, singular and plural, and tenses of verbs. I, therefore, recommend that authors double-check their grammar, ask their native English-speaking colleague to assist to check grammar, and/or use the English editing service to ensure that it is meet the standard of publication. Some of the observed errors will be mentioned through specific comments.
2) Conclusion, not a summary
The authors should rewrite the conclusion section. In the present form, only abbreviated results i.e. the main points are presented. A conclusion can include the summary of the main points, but it has to integrate the results and carry the message that the authors want to convey.
3) Figures
Generally, wherever the font and visibility of the image can be improved, it should be done. The data in figures is good, but I would suggest some modifications, you can see the details in specific comments.
Specific comments
1. I suggest changing the title into “Hydrochemical Characteristics and Water Quality of Shallow Groundwater in Desert Area of Kunyu City, Southern Margin of Tarim Basin”. Since you define the groundwater quality, which is based on the hydrochemical characteristics, the quality should come second.
2. Line 19 and 20: main/dominant – remove one
3. Line 20: what is TDS? You mention it here first, so it should be described.
4. Line 21: water chemistry types – change to hydrochemical types
5. Line 27: is - has
6. Line 89: reference missing for land use percentages
7. Line 95 and forward: reference missing for climate, precipitation and evaporation amount
8. Figure 1: increase the font where possible; uniform the font in legend; add graphical scale in two sub-maps; label the Duwa river in the map; what is the meaning of the red star in figure?
9. Line 104: references missing for lithological characterization
10. Line 106-107: loose rock pore water system? Do you mean intergranular porosity?
11. Line 123: surface slope of 16.70 ‰ is 1.67%, which is not relatively large.
12. Figure 2: increase the font for groundwater contour label and depth of water level in legend and in map; change the groundwater contour color, now it is blue label on blue background; increase the font of graphical scale. How did you obtain groundwater contour map? Which interpolation? Is data used on the same day? Low, medium, or high levels on groundwater? Please explain and modify accordingly.
13. Line 138: What is the meaning of 90 groups of samples? What is the sampling frequency? How many locations for 90 samples? Is every location sampled only once or some other sampling schedule was used?
14. Line 140: Where is Duval River on map?
15. Lines 141-143: Use past tense, e.g. the sampling bottles were washed etc.
16. Section 2.3. How did you obtain pH, TDS, TH, CO32- data? Please add text.
17. Line 165: can express
18. Line 201: cation content – cation order
19. Line 221-222: It can be observed (from the figure) that… The text in parenthesis not needed.
20. Lines 222-223: change sign for comma
21. Table 2: CODMn and EC are not explained anywhere in the text. Please add text.
22. Figure 3: increase the font in legend and in map
23. Figure 4: I understand the idea, but it's not very visible, try using a different color palette to see all the results.
24. Line 243: deser?
25. Line 251: Piper is a diagram, not a map. Please revise.
26. Figure 5: Change the font color to black to increase the visibility.
27. Figure 8: increase the font where possible
28. Figure 9: increase the font where possible; change the contour label color to be more visible; also, the contour color could be black, now it is green on green background
29. Line 364: it found?
30. Lines 370-376: the zone within the given classification are not unavailable, it is unsuitable zone. Also, the category is now good, but good to permissible. Be specific when describing.
31. Line 386: It is actually 5 samples, not 3. Please revise.
Reviewer 3 Report
It is a good paper.
The region should be presented in the title so the reader could understand the country of study.
I suggest you add a few quantitative results in the abstract.
The map should inform the study region is located in which country.
Please state which groundwater regions are suitable for drinking water.
I suggest you compare below references with your paper and add them:
A- Evaluation of groundwater quality using water quality index and its suitability for assessing water for drinking and irrigation purposes: Case study of Sistan and Baluchistan province (Iran)
B- A novel approach in water quality assessment based on fuzzy logic
C- Assessment of groundwater quality and evaluation of scaling and corrosiveness potential of drinking water samples in villages of Chabahr city, Sistan and Baluchistan province in Iran
Reviewer 4 Report
The study is well presented by the authors with aunthetic references and figures. I appreciate the authors. I have following two suggestions to be incorporated before being accepted as it is a case study.
1) Inclusion of results in abstract and conclusions
2) Typographical corrections at certain places
3) Location map can be changed to Global. Also the detailed map is required as many things are not visible.
4) Limitations of the study and future studies required can be added.
5) Sampling frequency is missing.
6) Land use details shall be given in detail.
7) Any improvement techniques possible can be suggested.
Reviewer 5 Report
This study evaluates the qroundwater quality in one of the cities in Tarim Basin by utilizing the water quality index (WQI) and irrigation water suitability related parameters. This study examines the conditions of the groundwater quality and provides guidance for future groundwater use whether it is suitable for irrigation and drinking water. The paper is well-written, and the figures and tables are appropriate. My recommendation is to return this review to the authors for minor revisions.
Comments to the Author(s)
Page 4 Line 160: Sodium percentage (SP) is mentioned in the sentence. Sodium percentage appears again in Table 1, Line 370, and Figure 11. I recommend to change sodium percentage (SP) in Line 160 into sodium percentage (Na%).
Page 7 Figure 4: The figure is describing the correlation between parameters in a very effective way by showing the correlation with values, colors, and scale. However, the values near 0 are hard to read due to light color. Would it be possible to slightly change the colors to make the values readable?
Page 7 Line 237: According to the cited reference (Miao et al., 2021), the Shukarev classification is based on six (seven?) ion concentrations and 23 ground water hydrochemical types. The sentence needs to be revised and needs provide more information for to clarify the difference between the ion concentrates and the hydrochemical types.
Page 7 Line 237: What are the seven categories in Figure 5? The legend in the figure shows six types.
Page 8 Line 262: Does the types described in the legend box in Figure 6 relates to the types in Figure 5? If so, it would be better to rename them with same names for better understanding.
Page 10 Equation (5) and (6): The right-hand-side of the equation contains C( ). What is C here?
Minor Comments
Page 1 Line 42: There are two periods in the end of the sentence.
Page 2 Line 77~78: ”… as the study area, Combined …” The comma should be a period or the whole sentence should be revised.
Page 3 Line 111: There is a double space in the sentence.
Page 6 Line 212 and Line 217: Add space between the value and unit. (3028.7 mg/L and 82.15 km2)
Page 6 Table 3: Add space between the value and unit.
Page 10 Line 297: Is this sentence describing Figure 3 or Figure 8(d)?
Page 11 Line 357: Adding the full name of USSL
Round 2
Reviewer 1 Report
---Add the R2 values on Figure 8.
---Check out the articles below.
ÇITAKOÄžLU, H., Çetin, M., Çobaner, M., & HAKTANIR, T. (2017). Mevsimsel yağışların jeoistatistiksel yöntemle modellenmesi ve gözlemi olmayan noktalarda tahmin edilmesi. Teknik Dergi, 28(1), 7725-7745.
ÇITAKOÄžLU, H., DEMİR, A., & GEMİCİ, B. (2021). REGIONALIZATION AND MAPPING OF DISSOLVED OXYGEN CONCENTRATION OF SAKARYA BASIN BY L‒MOMENTS METHOD. Mühendislik Bilimleri ve Tasarım Dergisi, 9(2), 495-510.
Reviewer 2 Report
Dear authors,
Thank you for the detailed changes, I have two more comments related to my remarks from the first round of review:
11) line 321 in the revised version of the paper - my point was that 1,67% is not a relatively large so please adjust this text
30) line 856 in the revised version of the paper - the category is good to permissible, not just good. Please revise
Author Response
11) line 321 in the revised version of the paper - my point was that 1,67% is not a relatively large so please adjust this text.
Response: A slope of 1.67% is too small, change this to: The ground slope of the floodplain area outside the southern part of the area is small, about 1.67ï¼…, and the aquifer particles are mainly sand gravel and gravelly silt.
30) line 856 in the revised version of the paper - the category is good to permissible, not just good. Please revise.
Response: ‘only 18% are in the good category’ has been replaced with ‘only 18% are in the good- permissible category’.